# Help-Seeking in Informal Family Caregivers of People with Dementia: A Qualitative Study with iSupport as a Case in Point

**DOI:** 10.3390/ijerph19127504

**Published:** 2022-06-19

**Authors:** Anna Messina, Rebecca Amati, Emiliano Albanese, Maddalena Fiordelli

**Affiliations:** Institute of Public Health, Faculty of Biomedical Sciences, Università della Svizzera Italiana, 6900 Lugano, Switzerland; rebecca.amati@usi.ch (R.A.); emiliano.albanese@usi.ch (E.A.); maddalena.fiordelli@usi.ch (M.F.)

**Keywords:** informal caregivers, iSupport, dementia, help-seeking, training interventions

## Abstract

Supportive measures and training interventions can improve the care of people with dementia and reduce the burden on informal caregivers, whose needs remain largely unmet. iSupport is an evidence-based online intervention developed by the World Health Organization to provide support and self-guided education to informal family caregivers of people with dementia. This qualitative study explored barriers and facilitators in the access and use of supportive measures for family caregivers of people with dementia living in Southern Switzerland (Ticino). We conducted five focus groups and explored experiences, beliefs, and attitudes toward seeking help (SH), and used thematic analysis to identify key themes. Participants (N = 13) reported a general reluctance to SH. We identified four main barriers to SH: high level of burden; sense of duty; fear of being misunderstood by others; and difficulty in reaching information. We also identified facilitators of help seeking behaviors and unveiled the need of caregivers to be assisted by a dementia case manager to facilitate access to support resources. Local services and interventions should be adapted to caregivers’ needs and expectations, with the aim of facilitating the acceptance of, access to, and service integration of existing and future support measures, including iSupport.

## 1. Introduction

Most people affected by dementia live at home [1] and are assisted by family members who provide instrumental support in daily living, and coordinate professional care delivery. Family caregivers of people with dementia are also defined as “informal caregivers” since they provide non-professional and un-paid care to help a person in long-term need [1]. Informal care is particularly prevalent in countries with a scarcity or lack of formal support services for people with dementia [2]. Being the primary caregiver of a person with dementia exposes informal caregivers to emotional, financial, and physical strain, and promotes the development of symptoms of psychological and physical distress, including depression, anxiety, loneliness, hypertension, and breathing problems [3,4,5,6]. Moreover, the care related burden in informal caregivers is positively associated with abusive behaviors, worsening of behavioral symptoms in the care recipients [7,8], and with institutionalization [9].

Multi-component interventions can have broad-ranging benefits in improving informal caregivers’ quality of life, symptoms of anxiety, and depression, and in reducing the care-related burden [10,11]. However, caregiver services and support interventions remain underutilized [12]. Ludecke and colleagues [13] showed that of 59,323 family caregivers who took part in a study involving 6 European countries, only 3% used support services. According to Brodaty [14] the main reasons for the low use of community services for informal caregivers of people with dementia living in Australia included a perceived lack of need of and reluctance in seeking help, despite self-reported high levels of burden and resentment. Other international studies found that high levels of burden, high levels of impairment in the care recipient, the lack of an informal support network, and fear of losing the role of primary caregiver were positively associated with a reluctance to use support services [15,16]. A systematic review aimed at exploring seeking help (SH) intentions and behaviors of family caregivers of people with dementia, reported the following barriers: inadequate knowledge about dementia, strong family norms about responsibility to caretake, stigma and bad experiences with health care services [17]. Most recently, an empirical trial [18] showed that family caregivers’ decision to reject support mainly depended on personal factors (e.g., caregiver gender or time); service factors (e.g., availability and knowledge) and relational factors (e.g., preferences of the care recipient). The authors also reported that a comprehensive assessment of caregivers’ unmet needs increased the rate of users willing to use the services. Overall, despite the growing number of support initiatives for informal caregivers, many barriers to accessing and using interventions and support services for caregivers still exist [14,19,20]. A better understanding of help-seeking behaviors and needs in family caregivers of people with dementia is needed to close the gap between the availability of support measures and their access and use, with potential benefits for the caregiver and the person with dementia [21,22].

iSupport is an online evidence-based program developed by the World health Organizations to provide self-guided education and support to informal caregivers of people with dementia [23]. The program includes problem solving and cognitive behavioral techniques such as psychoeducation, behavioral activation, and relaxation to support caregivers in everyday caring and in preserving their own well-being. More than 30 countries, included Switzerland, are currently involved in iSupport implementation worldwide [12]. Preliminary findings about iSupport are promising. Teles and colleagues [24] highlighted positive results of iSupport regarding caregivers’ knowledge in Portugal, while Oliveira [25] reported more positive attitudes towards the person with dementia in caregivers using iSupport in Brazil, compared to those who did not. In Switzerland, iSupport is currently being adapted in the Ticino Canton, the southern Italian speaking part of the country. Before being implemented, the program must be culturally adapted to local settings to ensure it meets the values, preferences and needs of the final users, as reported by WHO guidelines [26]. To the best of our knowledge, iSupport is the first online training intervention dedicated to the informal caregivers of people with dementia in Switzerland and there is no evidence regarding the local caregivers’ needs and/or acceptance of support interventions in general. Thus, we decided to use iSupport as case in point to investigate family caregivers perspectives about support measures and interventions, in order to collect evidence about the acceptance and potential use of support interventions.

As part of the iSupport cultural adaptation process, this qualitative study aimed to explore caregivers’ experiences, beliefs, and attitudes towards seeking help of family caregivers of people with dementia living in Ticino Canton. Specifically, we aimed to identify potential barriers and facilitators to the access and use of support measures, in view of adapting iSupport to caregivers’ local needs and enhancing its future use.

## 2. Materials and Methods

### 2.1. Study Design

We adopted a qualitative descriptive design [27] and used focus group as a data collection method to collect participants’ experiences and general attitudes towards support measures and the iSupport program. Focus groups provide insight into context-specific behaviors and beliefs and facilitate the expression of dissent and disclosure of sensitive issues, allowing the emergence of potential barriers and criticalities [28]. Methods and results are reported according to the COREQ checklist for interviews and focus groups [29].

### 2.2. Participants

We used a snowball technique to recruit a convenient sample of family caregivers of people with dementia. Eligibility criteria were: (1) being (at present or in the past) the primary caregiver of a family member with dementia, (2) being fluent in Italian and (3) living in Ticino Canton (southern Switzerland). Between April and May 2021, we designed an invitation letter and a flyer presenting the project, the main purpose of the focus group, eligibility criteria, and contact information. We disseminated these materials in a local newspaper, to members of the iSupport community advisory board, to the local Alzheimer’s association and day care centres for people with dementia, and to participants in ongoing research projects who consented to be informed about further research activities.

### 2.3. Study Procedure

Eligible caregivers interested in participating could contact us either via email or by phone; they then received a detailed presentation of the iSupport project, with examples of contents included in the program. The informed consent was sent via email at least 2 weeks before the data collection. Once we received the signed consent form, the participants were sent an online survey via REDCap (Research Electronic Data Capture) [30,31] to collect sociodemographic characteristics, and information about the degree of kinship with the person with dementia, the living situation of the person with dementia, and the length of experience in caregiving. Between June and August 2021, we conducted five focus groups at the Università della Svizzera Italiana in Lugano (southern Switzerland). Focus groups lasted approximately 2 h (between 110 and 130 min), and were audio recorded. One of the researchers (AM), a PhD student and psychologist with experience in the topic of dementia and caregiving, moderated all the focus groups, with assistance from members of the research team expert in qualitative methods (MF and RA). We adapted and expanded the WHO iSupport adaptation guidelines to our study aim, and we developed a topic guide composed of semi-structured questions. After a general introduction to the topic and an initial round of presentations, the discussion focused on three main areas. In the first part of the discussion we asked participants to express their experiences and attitudes towards support measures and training interventions; in the second part, we invited participants to evaluate the relevance and perceived utility of iSupport contents, and in the last part, we asked participants to provide feedback and suggestions to enhance the use of support interventions, including iSupport. For the purpose of this study, we focused on the first and third areas. The findings related to iSupport specific contents will be discussed elsewhere.

### 2.4. Data Analysis

Discussions were transcribed verbatim and pseudonymised by EB, an independent research assistant. AM, RA, and MF used thematic content analysis to identify key themes in different steps. First, researchers familiarized themselves with the data by reading and re-reading notes and transcripts to get an idea of the overall meaning and started to identify the core topics. Next, each researcher independently identified codes within each focus group (vertical analysis) and across the whole dataset (horizontal analysis) to find variations and regularities within the data. Themes and subthemes were progressively refined through discussion in weekly meetings over three months (from November 2021 to February 2022), and until consensus was reached. Data management and coding processing were facilitated by the use of NVivo12 software. We tracked and stored all researchers’ reflections and decisions that were made throughout the data analysis process on an online shared document, as a basis for a meta-reflection about the inductive process.

### 2.5. Ethical Issues

We received the right to proceed from the Swiss Cantonal ethics committee, since our study did not fall within the scope of Art. 2 and Art. 3 of the law on human research and did not require any ethical approval.

## 3. Results

The results are structured as follows. First, we present the socio-demographic characteristics of the participants. Second, we report the overarching themes including the barriers and benefits related to seeking help, stratified by professional and informal forms of help. Third, we describe the virtuous process between positive support experiences and seeking help behaviours, and a profile of an effective the dementia case manager is created, as defined by participants through discussion. All results are supported by comments, translated verbatim, and marked with a number indicating the participant’s ID and the type of relationship to the person with dementia. Quotations were first translated from Italian to English by a member of the research team (AM) and then revised by an external translator.

### 3.1. Sociodemographic Characteristics

Of the 20 caregivers who contacted us, 13 participants joined the focus group discussions. Reasons for withdrawal were lack of time and geographical distance. Most caregivers were female (N = 10), and their ages ranged from 55 to 82 years. Their care recipient was either a parent (N = 6) or the spouse (N = 7) of the caregiver. More than half of participants (N = 8) reported to have taken care of the person with dementia for at least the past 3 years. At the time of data collection, most caregivers cared for a family member with dementia living at their own residence (N = 8). Two participants reported the person they cared for had passed away.

The number of participants attending each group discussion ranged from 2 to 7 caregivers. Focus groups were attended by 6 caregivers (FG1: 14.06.21); 7 caregivers (FG2: 12.07.21); 2 caregivers (FG3: 15.07.21); 6 caregivers (FG4: 18.08.21), and 2 caregivers (FG5: 24.08.21), respectively. The majority of participants attended more than one group. Table 1 shows the number of focus groups attended and sociodemographic characteristics of the 13 family caregivers who took part to the focus groups.

### 3.2. Barriers in Seeking Help

The main emerging theme from our analysis was a general difficulty in seeking help, defined as “the active process of seeking help” [32]. In most cases, caregivers expressed the need to be supported, but faced barriers that prevented them from actively seeking and accessing both professional and informal forms of support. In our groups, professional or formal help included seeking assistance or advice from health and social care professionals including family doctors, physicians, psychologists, lawyers, and health workers. Informal or non-professional help referred to the process of seeking support at the community level, including family members, close acquaintances, and especially from other caregivers of people with dementia.

#### 3.2.1. Feeling Burdened

For the majority of caregivers (N = 7) the difficulties in balancing caregiving duties with other life and work demands represented a main barrier in seeking help. Specifically, some participants acknowledged that they needed and would benefit from support but highlighted how feeling stressed and overwhelmed prevented them from gathering information about and navigating existing services, and actively accessing and using support services. For instance, one participant used the metaphor of “drowning” to describe the feeling of burden that prevented her from even realizing the need of seeking help:


*Look for help… sometimes you don’t even think about it… you’re so absorbed, running here and there, and you’re so much drowning, you would really need someone there to tell you “come on take a break, hold still, I’ll take care of it now” but you (on your own), you don’t even get to look for help.*

*(P.5, spouse)*


For another participant learning about other family caregivers’ experiences led to additional emotional stress:


*Sometimes I said to myself, “I’ve got enough on my plate with my own problem.*

*(P.7, spouse)*


As a counterevidence to this, P.12 brought her experience of feeling less burdened a few months after her father’s diagnosis; therefore, being willing to share positive and negative emotions with other family caregivers:


*At the beginning I felt so bad, I didn’t even have the strength to react, and now that I feel better it would be important for me to meet people and share these (experiences)… of helplessness, of resignation, of excitement, and to talk about the happy moments.*

*(P.12, daughter)*


#### 3.2.2. Sense of Duty

Feeling caregiving as a family and personal responsibility contributed to preventing participants from seeking support elsewhere, especially if the relationship was between children and parents. Fulfilling caring duties alone was a common experience among participants, and was often considered as a personal responsibility, or even a moral obligation. However, one participant pointed out how considering herself as the main and sole caregiver may also lead to isolation and jealousy towards other caregivers:


*Many times, I’ve seen people isolating themselves, craving to be the one and only shouldering the burden… then it becomes some sort of jealousy “no one else is as good as me”.*

*(P.9, daughter)*


The strong sense of dedication in taking care of the person with dementia was so strongly felt by some participants as to be represented through epic metaphors, where the caregiver described himself as the “hero” and caregiving as a “mission”:


*But I think I’ll know when it comes the time to decide (nursing home admission), and that I don’t have to be the hero at all costs.*

*(P.12, daughter)*



*If you decide to take care of someone, you practically dedicate your whole life to this person… you must find the time to tell yourself “I’m going on because this is my mission, and I have to accomplish it”*

*(P.7, spouse)*


And yet, P.7 remarked that shouldering caring responsibility alone had benefits regarding self-esteem and pride, and provided with a sense of accomplishment.


*Maybe it was also a matter of self-esteem “I have this problem with my wife, and I have to fix it.”*

*(P.7, spouse)*


#### 3.2.3. Feeling Misunderstood by “Others”

Caring for a person affected by dementia was described, fairly unanimously, as a unique experience. A common belief among participants was that people who did not take care of a loved one living with dementia could not truly understand their needs and experiences. Therefore, caregivers preferred not to seek help and relied on their own resources, as shown in the extract below:


*Sometimes it occurred, unhappily, that when I was talking with people which have never experienced such a situation (living with a person affected by dementia), they would unintentionally put me off instead of supporting me, telling me “you have to be patient… “ they made me feel worse…so I decided to give up talking) with other people about it.*

*(P.12, daughter)*


Caregivers also reported that externals, either experts, friends, or relatives, tended to minimize the severity of the disease or caregiver’s distress. For example, P.9 brought feelings of sorrow and loneliness after having shared her experience with other people:


*Sometimes it can be very painful because you feel alone and so laden down… besides, other people minimize (your effort) or make comments that only confirm that you are alone and can’t rely on anybody to get by.*

*(P.1, spouse)*


#### 3.2.4. Difficulty in Attaining Information about Available Services and Support

Most participants reported to feel disoriented or even lost, especially at critical stages of the disease when support in decision making and problem solving were most needed, including after the disclosure of dementia diagnosis, and as symptoms evolved and worsened through the course of the disease. Caregivers looked for different kinds of information, namely assistive domestic aids, financial support measures and medical assistance for the person with dementia. P.5 for instance reported great difficulties in finding a professional domestic helper:


*At some stage we needed a person (a formal caregiver)… I spent two months searching … I even asked a friend of mine who worked in a cleaning company, and I asked her “listen, do you have someone who is really brilliant?” I was really at the end of the rope.*

*(P.5, spouse)*


P.8 needed adapted kitchen appliances for the safety of his father living with dementia, but she could not find an existing, competent provider, and eventually had to find a work around by herself:


*(My father) had been turning on the heat and burning everything…I asked the electrician, and he didn’t even know where to start…it took me a week to solve it.*

*(P.8, daughter)*


Some participants also highlighted a general lack of integration of, and coordination among local dementia support services that made it cumbersome for them to locate and make good use of relevant information. For instance, P.6 encountered puzzling barriers to access information about available economic support measures for people with dementia and their families:


*To know how it works with disability, with the pension fund…Honestly, there’s a lot of confusion… what’s more, it’s quite difficult to get in touch with them… before you get an answer, it goes on for a long time…You go here you go there and nobody knows exactly how it works.*

*(P.6, son)*


Similarly, the navigation of existing services proved taxing for most participants. For example, one participant pointed out the disruption of local services, and suggested that a sole entry point, reference service or institution should be available to provide timely and appropriate responses to caregivers, coherently and comprehensively:


*This is a great trouble, there is no service you can call or turn to, and that they tell you “sir, madam, you are entitled to this and that.”*

*(P.1, spouse)*


Because the dementia diagnosis was somewhat unexpected, and provoked confusion for both caregivers and care recipients, participants expressed the need for a structured, yet tailored approach to the disclosure of a dementia diagnosis, associated with the provision of key information of the main implications of diagnosis, the associated needs, and existing services and organizations, and how to approach them in a timely manner. P.6 perfectly expressed the impact of the diagnosis and the feeling of being “abandoned”:


*You feel really abandoned because there is no organisation behind it… Even because you have to deal with a person who used to do all on his/her own, just a month before or a year before, and now is completely lost.*

*(P.6, son)*


### 3.3. Benefits of Seeking Help

Despite the above-mentioned barriers in seeking help, some participants also reported positive support experiences, which differed according to whether they sought professional or non-professional help. Professional help mainly consisted in care provided by family doctors or health care workers. Non-professional help was informal, and it consisted mostly in seeking support from other caregivers in the community through peer-to-peer interactions. These benefits are presented, starting with professional help experiences.

#### 3.3.1. Ensuring the Safety of the Person with Dementia

Participants concurred that one of the main advantages from seeking professional support was the achievement and improvement of the safety for the person with dementia. The decision to seek support from professionals often resulted from a worsening of symptoms in the care recipient and consisted in hiring a domestic worker or asking advice from a specialist or family doctor. For P.3, for example, the main concern was leaving her father alone at home and the risk of domestic accidents:


*There is a paid caregiver coming in the morning and in the evening because I can’t leave my father alone … he can fall… he broke his femur at home several times, and I don’t feel comfortable to leave him alone… but I don’t want to put him in a nursing home… not yet let’s say.*

*(P.3, daughter)*


Other participants also emphasized the benefits of therapy and medications for the person with dementia, in a manner perceived as safe and accurate:


*Even medicines have to be administered by someone external, if necessary, because she (the mother) used to tell me “Yes I took them” then once downstairs you could see she didn’t … she finds all the excuses, there must be an external person who has the control.*

*(P.8, daughter)*


For another participant professional support contributed not only to improved safety, but also to improved quality of life for the person with dementia:


*On doctor’s advice I found a paid caregiver who visits him four or five hours a week… it’s nice because they’ve created a nice relationship… she takes him out for a walk or to the lake, they do nice things together, and at the same time she helps him to talk, she brings out many problems … it’s a nice thing.*

*(P.5, spouse)*


Some participants stressed the importance of accessing formal support as early as possible, with respect to the clinical diagnosis of dementia and the onset of even mild symptoms. Timely support from professional caregivers can help to anticipate needs, formulate directives, and plan for mid- and long-term care whilst the care needs of the person with dementia are still relatively limited and manageable. For example, P.6 reported:


*The sooner the support arrives, the better … I experienced it, and at a certain point you have no chance… you must seek help…you must provide care that is not only love, but medical, domestic, physiotherapeutic care. *

*(P.6, son)*


#### 3.3.2. Relief

Seeking professional support and sharing caregiving duties not only brought benefits to patients, but also led to a feeling of relief for the caregiver. Participants reported how accessing respite measures enabled them to find time and energy for themselves and provided a sense of reassurance, which contributed to reduced anxiety and distress. Respite measures included for example day-care facilities where the person living with dementia can spend time doing a variety of activities while the caregiver may take a break from caregiving duties. P.2 significantly described this as “salvation”:


*My mum went to a day care centre, and she has been there for seven or eight years now …they also involved her in the cooking… Fortunately there was that salvation twice a week… otherwise things are too long to be maintained.*

*(P.2, daughter)*


However, for other participants, relief was actually perceived only after admission to a long-term care facility. Despite some initial reluctance about institutionalization, caregivers reported that institutionalization contributed significantly to feeling released and “cared for”:


*This (admission to a nursing home) was good for her, and good for me as well because I felt cared for, and this is a great support… it helps to feel helped by someone.*

*(P.7, spouse)*


According to P.9, sharing care responsibilities and accessing support measures also helped her to resize her role as caregiver and to see the disease itself in a more manageable way:


*If you use all the supporting measures that exist, I wouldn’t say that the problem becomes small but… after all they are experts in this.*

*(P.9, daughter)*


#### 3.3.3. Free to Speak

The interaction between caregivers as a form of reciprocal informal support provided the opportunity to express feelings and worries and to share thoughts about the caregiving experience without being judged. Differently from external supporters, such as relatives or friends, other informal caregivers were perceived as more understanding and empathic because they shared the same caregiving experiences and faced similar or even identical issues and personal concerns:


*If you have never experienced caregiving, you judge…but if you have been through it, you won’t judge.*

*(P.12, daughter)*


For P.9 sharing the common experience of caring for a person with dementia also facilitated the expression of the challenges and negative emotions that the caregiver may feel towards the care recipient, including anger and guilt:


*You realize that also other people have similar feelings… because you feel guilty when you go crazy, and you feel bad … you know it is wrong because you don’t do it on purpose, but you realize that also the others (caregivers) do it (to get angry) … people understand you because they are experiencing the same situation … when people who don’t live this tell you “yes I understand” I am convinced they can only partially understand *

*(P.9, daughter)*


#### 3.3.4. Sense of Belonging

The freedom to share both positive and negative aspects of caregiving in a non-judgmental environment, also generated a sense of belonging to a group. Caregivers reported to feel less lonely and stressed after having interactions with peers. According to P.12, interactions with peers provided unsought opportunities to appreciate serious situations and problems to which they could compare their owns:


*I’m surprised people don’t join help groups… because there you realise that you’re facing difficulties but hearing other people talking about more serious situations… you go home and you have recharged your batteries a bit.*

*(P.12, daughter)*


For P.2 the level of intimacy experienced with other caregivers, turned the self-help group in a group of friends with whom they could share not only caregiving experiences, but also other matters, and with whom they could enjoy some free personal time:


*You realise you are not alone…other people have the same problems you have and talking about that helps you … you realise some reactions you have are shared by other people, and this reassures you a bit… it heartens you. Moreover, we hang out together and we became almost friends, we meet for a drink, and we cry, we laugh.*

*(P.2, daughter)*


#### 3.3.5. Problem Solving

Different from professional help-seeking, sharing and listening to family caregivers’ experiences fulfilled the need for emotional support and nurtured a sense of belonging to a group. Participants also reported problem solving as an additional benefit from informal help-seeking. Peers provided practical forms of support, such as help in gathering information or making decisions concerning the person with dementia based on their lived experiences. Moreover, for some participants, the interaction with other family caregivers satisfied their personal need to confirm that they were taking care of their loved one in an appropriate manner and highlighted potential improvements, as reported by P.2:


*You get together with people who have the same problems … because you don’t know if you are doing something wrong…you don’t know anything, and personally it really helped me because there is an exchange even in simple things, in everyday practical things, and we feel part of it.*

*(P.2, daughter)*


In addition, interacting with “experienced” informal caregivers also helped participants to learn strategies and ways of coping with every-day challenges, turning the interaction into a mutual learning process. In this regard, speaking of the reason she joined the focus groups, P.1 said:


*We must address people who give us help and support, and who have more experience, that is … I answered the question “why did you decide to participate?” like this: to be informed and to learn from those who are at an advanced stage in this experience.*

*(P.1, spouse)*


Sometimes, participants even attributed more knowledge, expertise, and availability too peers than to professionals who worked in the field of dementia. For instance, P.6 reported:


*These family members all together know much more (than the experts), and are also more willing to talk…you see the doctor for five minutes and then you go…*

*(P.6, son)*


### 3.4. The Virtuous Cycle of Seeking Help

Despite the difficulty experienced in seeking help for the first time, caregivers reported how experiencing the benefits mentioned above from both professional and non-professional support, helped change their initial help-seeking attitudes and facilitated the maintenance of help-seeking behaviors over time (Figure 1).

P.2 provided an accurate account of her initial skepticism about joining a self-help group with other family caregivers and how then she changed her mind:


*Concerning the self-help group, I came to know about it through my sister who proposed it to me… honestly, I told her “but yes… I mean, it won’t do any good”…I was very sceptical, and then I realised that it was really useful in our case… we have been participating regularly for two years now.*

*(P.2, daughter)*


Some caregivers reported the importance of seeking help from a professional who was not only expert in the field of dementia, but who was also in touch with people living with dementia. This probably helped them to overcome the fear of being misunderstood by others and to start seeking support, as shown in the extract below:


*I heard that there was also a psychologist, so I approached her, and I thought it could be useful … but I didn’t want a psychologist, let’s say a generic one, I wanted someone who worked with patients … so they gave me a name, and now I see her once a month to talk.*

*(P.1, spouse)*


The access and repeated use of professional care services also depended on the benefits experienced, not only by the caregiver, but also by the person living with dementia. For instance, P.8 brought the experience of her mother who attended supported holidays for people living with dementia:


*We’ve always sent her on assisted holiday both to the seaside and in the mountains …they go every year to get to know each other… she enjoyed it a lot because they took her on trips, and even now she’s looking forward to go to the mountains, and then afterwards she’ll be waiting for the Christmas party… there will be always something to wait for…*

*(P.8, daughter)*


### 3.5. The Dementia Case Manager

Participants often referred to different needs pointing to a possible solution to overcome barriers in help-seeking behaviors, in particular the provision of a reference person to help them navigate dementia care services and manage caring duties. Caregivers identified three main features that this putative dementia case manager should have and that pertained both to professional and non-professional forms of help: knowledge of the family and caring situation, knowledge of local resources and services for dementia, and long-term availability (Figure 2).

#### 3.5.1. Knowledge of the Family and Caring Situation

According to caregivers, a dementia care manager should be reliable, expert in the field of dementia, but also familiar with the person with dementia and their needs, resembling an “extended” member of the family. Knowledge of the caring context and contact with the person living with dementia were essential requirements for caregivers to receive tailored and need-centered support, as reported by P.8:


*It is essential to rely on someone… a person you can count on… if he’s not part of the family he may be external, but someone you can really count on… not a doctor… someone who follows you as well as the person you take care of.*

*(P.8, daughter)*


#### 3.5.2. Knowledge of Local Resources

Caregivers reported the need to address someone who knows what local resources are available for people dementia and to advise on which of these may be most suitable for them. Some participants stressed the importance of being able to refer to a single person, as a means of continuity of care and to find information and solutions for the person they cared for through all phases of the disease, promoting integration and coordination of care:


*He should come from outside … we need the experience of someone who gathers all the available resources, and puts them in a network.*

*(P.6, son)*


#### 3.5.3. Long Term Availability

Participants pointed out that support should be provided not only at the beginning, but throughout all phases of the disease, to help the caregiver to make decisions at different stages of the caregiving journey. Rather than a consultation from time to time, caregivers reported the need for continuous support and guidance, as expressed in the extract below:


*I believe that a common thread is missing… someone who can stay there from the beginning to the end with advice, who knows how things work… But here there is no one at all, that’s why there is confusion.*

*(P.5, spouse)*


Long term availability and contact with the person with dementia and his family are the elements that most distinguished the profile of the dementia case manager from other health professionals, as expressed by P.8:


*We would need someone who also knows mum and sees how she is, and what she needs in different moments… yes, the doctor… they are all very nice… the geriatrician maybe takes a bit more time, but he visits her for an hour every six months… we would need a contact person who can discuss with us, the relatives, who sees how she goes on…who is also there for us to solve practical problems.*

*(P.8, daughter)*


More specifically, P.6 compared the role of a “dementia case manager” to that of an “architect”. This suggests the need of having a professional who builds individual health-care plans based on the necessities of the person with dementia and his caregivers:


*There is not even a professional figure … if I have to build a house I go get an architect.*

*(P.6, son)*


According to P.1, besides providing support in the organization and coordination of care, a dementia care manager may also contribute to reduce feelings of burden, uncertainty and loneliness, by “holding hands with” the caregiver throughout the progression of the disease:


*There are people who feel insecure and overwhelmed because of the terrible reactions the person with dementia may have… you must get used to it, but you don’t have so much energy because you are already busy to manage it … I am sorry because “to manage” is not a nice expression, but I didn’t find another one because you can’t say “assist”… you really have to “manage it”… So, his role should be taking them (the caregivers) by the hand and showing what in that specific moment and for that specific situation can help.*

*(P.1, spouse)*


## 4. Discussion

In this study we explored the experiences, beliefs, and attitudes towards seeking help in family caregivers of people with dementia living in southern Switzerland (Ticino). We found a general reluctance for seeking help, despite reported feeling of burden and stress due to caregiving demands. High levels of burden, fulfilling a sense of duty, feeling misunderstood by others, and difficulty in reaching information about available services were the main barriers to seeking both professional and non-professional forms of support. Caregivers also reported some benefits from support experiences including safety for the person living with dementia, emotional relief, a sense of belonging, and the freedom to speak about personal experiences in peer-to-peer interactions. Participants reported the need to refer to a dementia case manager who knows the person with dementia, their situations and needs, and who can provide caregivers with continuous support, guidance, and assistance to navigate and facilitate access to local services.

### 4.1. Barriers and Benefits to Seeking Help

Overall, our results are consistent with previous studies that found a general reluctance from informal caregivers of people with dementia to seeking help [11,13]. Similar to the work of Zwigmann and colleagues [18], we found that the decision to reject different forms of support depended either on personal (burden), cultural (sense of duty), relational (feeling misunderstood by others), and environmental (difficulty in reaching information) factors. In our study, high levels of burden hindered caregivers from actively seeking support. A recent work aimed at examining the relationship between perceived help-seeking difficulty and burden found positive associations between caregiver self-criticism and seeking help, but not with burden [33]. Conversely, a review showed that high levels of burden and poor health were not only associated with a reluctance to use support services, but also with a poor knowledge of the services available for caregivers [11]. Contrary to common thinking that help-seeking is primarily dictated by need, there are, in fact, complex processes that involve different and sequential steps of decision making, starting with the recognition of needing help in the first place [34]. The feeling of being overwhelmed and “drowning” (as described by one of our participants) in a usually unexpected role, may prevent caregivers from realizing the need to be supported.

Some participants in our study remarked that providing care to a family member with dementia is a moral responsibility, or a duty. The influence of cultural values and family norms on help-seeking behaviors is quite well established in the literature, and varies across countries [35,36]. A recent qualitative study [37] found that Chinese cultural belief of filial piety, the filial obligation to provide for and look after elderly parents, played a main role in coping strategies adopted by family caregivers of people with dementia. Partially consistent with our results, authors found that although it prevented them from from seeking support outside the family, filial piety motivated caregivers to accept their caring role and to adjust themselves to daily caring duties. Indeed, besides contributing to isolation, participants in our study remarked how shouldering caring responsibility provided caregivers with a sense of accomplishment and pride. However, even if dedication and responsibility may enhance acceptance and motivation to provide care, there is also evidence that caregivers motivated by a sense of duty, guilt or social norms are more likely to progressively isolate themselves and suffer from psychological distress [38].

Another barrier we found to help-seeking was the feeling of being misunderstood by others, especially relatives or friends. Previous studies found that family caregivers who minimized or denied dementia symptoms [39], or experienced stigma [40], were less prone to seek help. However, there is no evidence regarding how social denial or minimization influence help-seeking behaviors. Our findings highlight the importance of attitudes towards dementia and caregiving of people not directly implicated in care provision and may have potential implications to ameliorate dementia awareness and prevention.

Finally, caregivers reported general difficulties in reaching information preventing them from accessing support when needed. This finding is consistent with the adapted Health Behavioral Model for family dementia caregivers [18,41] that defines availability and accessibility of services as essential facilitators of help-seeking behaviors in caregivers. Similarly, a recent study [33] found that difficulties in help seeking were largely related to the caregivers’ perception of services as complex and somewhat abstruse and, thus, inaccessible. Our results confirm the importance of providing not only useful support resources, but also clear, comprehensive, and structured information on how to reach, access, and navigate these resources efficiently and effectively.

In addition to barriers, participants also reported benefits from positive support experiences. More specifically, seeking professional support, mainly consisting of consulting a doctor, hiring a formal caregiver, or accessing health-care facilities, contributed to instill a sense of increased safety for the person living with dementia and a feeling of emotional relief for caregivers. There is increasing attention being given to barriers and facilitators associated with seeking help behaviors [17,33], but evidence is thin on the benefits experienced by family caregivers who access and use support programs and measures. A qualitative study explored the perspective of family caregivers, and their beliefs and motivations to use respite services, and found that the safety of the care recipient was a primary reason to use services, but only in case of functional deficits [42]. Similarly, a cross-cultural study reported that reasons for using support services included worsening in the condition of the person with dementia and the services’ ability to meet his needs [43]. Our results confirm that support conveys respite and relief to caregivers, but also suggest that caregivers may only reluctantly admit seeking help for themselves rather than for the person they care for [44].

Participants also reported benefits from reciprocal informal support, namely from interactions with other family caregivers. The opportunity to speak about personal concerns without being judged, and the sense of belonging to a group were among the main advantages reported by caregivers. In accordance with the expectation of being misunderstood by others, this finding remarks on the importance for participants to sharing their personal experiences with other family caregivers. Peers’ interactions nurtured the sense of belonging to a group whose members were considered trustful and non-judgmental. According to social psychology theories, sharing a common fate shapes a group’s identity, and delineates its boundaries [45].

Peer-to-peer interactions also provided opportunities to learn new coping strategies to deal with practical caring problems. Caregivers reported that peer learning helped them to develop a better confidence in their role and satisfied their need for confirmation. Indeed, low levels in self-efficacy were found to facilitate seeking help behaviors [39].

### 4.2. The Virtuous Cycle of Seeking Help

Even though our study identified several barriers that prevented informal caregivers from seeking support, we also unveiled benefits of professional and non-professional support experiences, which facilitated the maintenance of help-seeking behaviors over time, triggering a virtuous cycle of seeking help. Different theoretical models posit the existence of three main dimensions in help-seeking: attitudes and beliefs, intentions or willingness, and actual behavior [17]. Among these, an extended version of Andersen’s Health Behavioral Model [41,46] identified four relevant dimensions for the use and non-use of services from family caregivers of people with dementia: service factors (e.g., availability, accessibility, and cost); personal factors (e.g., health belief, needs); experiential factors (e.g., caregiver burden, clinical characteristics of the person with dementia); and relational factors (e.g., relationship with the care recipient). Our results may expand the evidence on the importance that past positive support experiences have in maintaining help-seeking behaviors and overcoming potential barriers, specifically through the change of negative attitudes and beliefs.

### 4.3. The Dementia Case Manager

Caregivers also expressed the need of being supported throughout the progression of the disease by someone who had expertise, but also knowledge of the family context and of local resources, namely a “dementia case manager”. According to participants, the dementia case manager may mediate between caregivers’ specific needs and support services, by “holding hands” with the caregiver through their caregiving journey. In public health, the implementation of case managers is a recognized means to promote an integrated and person-centered care approach [47]. Instances of application exist for diabetes and other non-communicable diseases, in which care needs and treatment require both self-care and long-term interaction with health and social care providers and services [48,49]. Preliminary evidence suggests that dementia case managers should have a professional background (nursing or social work) and interpersonal skills. The perceived benefits of case managers included the expected provision of practical and emotional support and facilitating access to health and social care services [50]. A recent review [51] found that, if in collaboration with family physicians and health care services, case managers can have a pivotal role in addressing the needs of patients and their informal caregivers. Nonetheless, despite promising results, case management interventions are rarely implemented, particularly in North America and Europe [51] or systematically evaluated [52]. Our results suggest the potential benefits of integrating dementia case management into the care pathways for people with dementia and their informal caregivers in Ticino Canton.

### 4.4. Implications for the Context and iSupport

Southern Switzerland (Ticino) has a health policy and legislation that aim to keep dependent older adults at home for as long as possible, offering financial support to informal caregivers to provide homecare [53,54]. In line with the national guidelines [55], in 2016, the government released a dementia strategy plan to improve the quality of life of people with dementia and their caregivers. Action areas include the improvement of dementia awareness and the extension and integration of care-paths for people with dementia and support resources and services for caregivers [56]. Caregivers can rely on different forms of support (i.e., local homecare service providers, respite services, and mutual support groups). However, resources are still poorly integrated, difficult to access, and underused.

We identified several barriers that deter family caregivers from seeking help and support. Our results can inform the re-centering of supporting measures on family caregivers’ needs, values, and preferences, and can contribute to improve the accessibility, acceptability, integration and, where needed, the development, and implementation of interventions including iSupport. Indeed, we found that, independent from the nature and contents of iSupport, caregivers expressed reluctance in using and assessing support interventions.

We believe that considering caregivers help-seeking behaviours and experiences as part of the iSupport cultural adaptation process is crucial to enhance its use and acceptability. For instance, caregivers reported to having a personal and moral duty towards the loved one for whom they provide dementia care. Therefore, iSupport should aim to integrate the care provided by caregivers rather than replace it, acknowledging the pivotal role of informal caregivers. Further, if we consider caregivers difficulty in navigating information about support services, we may ensure that iSupport becomes not only a place to exchange knowledge about dementia, but also a vehicle to acknowledge and access other forms of support and facilities. Additionally, the numerous benefits reported by participants from peer-to-peer interactions, such as the chance to speak freely, the sense of belonging and problem solving, suggest the importance for caregivers to share experiences with other informal caregivers. iSupport may address this need by integrating more interactive functions, such as chats or forum sessions where participants can engage not only with exercises but with other participants. Lastly, the reference to a dementia case manager and its features (knowledge of the caring situation; knowledge of local resources and services for dementia; long term availability) suggested to us the importance that this role may have in supporting family caregivers in being informed in using iSupport. Hence, the involvement of both formal and informal caregivers in iSupport’s adaptation and implementation process seems to be crucial to overcome acceptance and usability barriers. All these considerations apply to how iSupport is locally adapted and implemented, fulfilling its promise to meet informal caregivers needs, and to become a commonly and widely used resource, with local salience and pertinence.

### 4.5. Limitations

Some limitations are worth noting. First, our findings are limited to a small sample size and may be extended to other contexts only in part. The discrepancy in group size across focus group discussions, and the attendance of participants in more than one group, may have contributed to rapidly reach data saturation. However, group compositions varied within groups, and we found that this contributed to reduce other biases such as the social desirability effect. Second, our research focused on a narrow and specific cultural and health care setting in Switzerland, and family caregivers were not very heterogeneous in terms of gender, nationality, age, and living situation. However, we decided to use qualitative methods to elicit and explore in depth the nature of barriers in accepting and using iSupport at the local level. Similar approaches are warranted in all settings and contexts where iSupport is being implemented. Moreover, our results were highly consistent with previous evidence on caregiving experiences. This suggests that other family caregivers’ needs may be similarly unmet, and analogous barriers may exist in other contexts. Third, we did not consider important factors that may have influenced help-seeking attitudes and experiences during data collection, such as the severity of symptoms of the person with dementia or the access to support services. However, we did not aim to identify potential causes and facilitators to seeking help behaviors, but rather to give voice to caregivers’ lived experiences and needs. Lastly, we explored seeking support as the general process of actively seeking help, without distinguishing between the single stages included in the process (e.g.,: awareness and identification of the need; procurement of resources; and communication with others) or between the specific type of help sought (e.g., seeking professional consultation; use of respite service; access to psychosocial interventions) [17]. However, in the process of data analysis, we differentiated between professional and nonprofessional help to reveal the differences experienced by participants in terms of benefits, while this distinction was not required for the barriers.

In summary, this study provides a novel contextualized understanding of needs, beliefs, and barriers in help seeking behaviors of family caregivers of people with dementia in Switzerland, which should be considered when developing and implementing support measures, including locally adapted versions of iSupport.

## 5. Conclusions

In this study, we found that informal family caregivers of people with dementia living in southern Switzerland were reluctant in seeking help, and that several barriers exist that may deter the access and use of support measures and interventions, including iSupport. We also found evidence that positive support experiences reinforce the maintenance of help seeking behaviors over time. Local policies and dementia services should be adapted to account for the perspectives, values, preferences, and actual needs and expectations of caregivers, with the aim of facilitating the acceptance, access, and integration of existing and future support measures, including iSupport.

## Figures and Tables

**Figure 1 ijerph-19-07504-f001:**
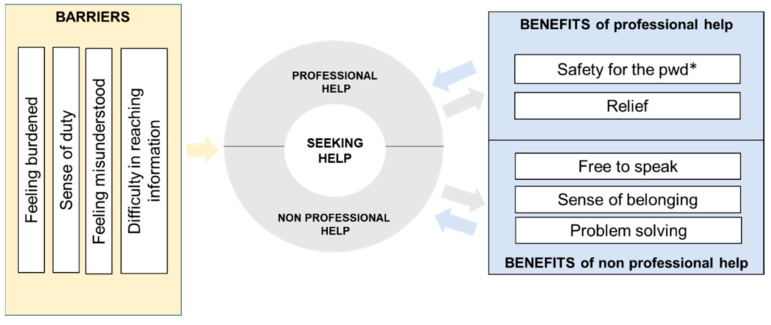
The virtuous cycle of seeking help. * person with dementia.

**Figure 2 ijerph-19-07504-f002:**
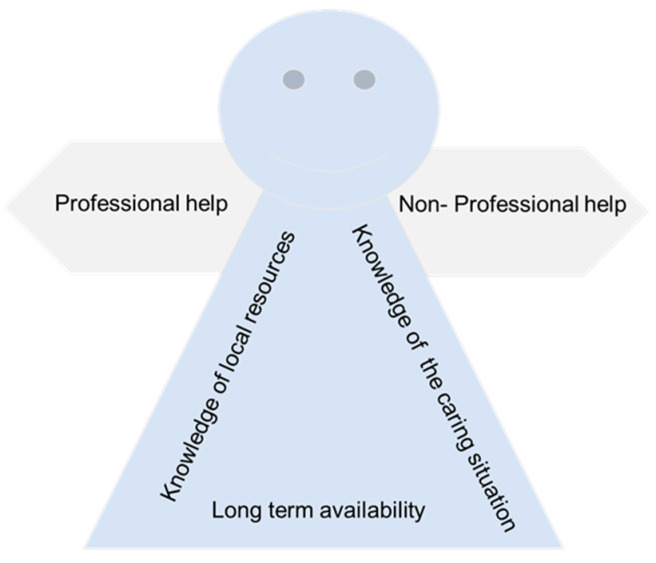
Profile of the dementia case manager.

**Table 1 ijerph-19-07504-t001:** Sociodemographic characteristic of participants.

ID	Gender	Age	Employment Status	Relationship with the pwd ^1^	Living Situation of the pwd ^1^	Years of Caring Experience	The pwd ^1^ Has Passed Away	Focus Group Attended ^2^
1	Female	58	Housewife/ Retired	Spouse	Own residence	3–5	No	5
2	Female	55	Housewife/ Retired	Daughter	Own residence	3–5	No	1; 3; 4
3	Female	59	Housewife/ Retired	Spouse	Own residence	3–5	No	1
4	Male	67	Employed	Son	Own residence	3–5	No	1
5	Female	58	Housewife/ Retired	Spouse	Caregivers’ residence	1–2	No	2
6	Male	57	Employed	Son	Own residence	3–5	No	1; 4
7	Male	74	Employed	Son	N/A	6–10	Yes	1; 2
8	Female	55	Employed	Daughter	Own residence	3–5	No	2; 4
9	Female	75	Housewife/ Retired	Daughter	N/A	>10	Yes	1; 2; 4; 5
10	Female	76	Housewife/ Retired	Spouse	Own residence	3–5	No	2
11	Female	82	Housewife/ Retired	Spouse	Caregiver’s residence	1–2	No	2
12	Female	55	Employed	Daughter	Own residence	1–2	No	3; 4
13	Female	81	Housewife/ Retired	Spouse	Caregiver’s residence	3–5	No	2; 4

^1^ Person living with dementia. ^2^ Number of the focus group discussion attended. N/A not applicable.

## Data Availability

Extracts from the datasets used during the current study are available from the corresponding author on reasonable request.

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
