# Peer review of "Help-Seeking in Informal Family Caregivers of People with Dementia: A Qualitative Study with iSupport as a Case in Point"

_ijerph, 2022, doi:10.3390/ijerph19127504_

Round 1

Reviewer 1 Report

Thank you for your innovative work with this unique data set and country context. Overall very well written. I have offered a number of detailed suggestions to further strengthen your article, for your review and consideration. I look forward to reading your major revision. The following suggestion:

Title

A precise title is "Experience of Seeking Help: Using a Qualitative Study for the Informal Family Caregivers of Patients with Dementia".

Term

Please check the correct use of acronyms. They must be expanded at the first occurrence of the acronym. For instance,"supportive measures and training interventions can improve care of people with dementia and reduce the burden in informal caregivers, whose needs remain largely unmet." and the others "Most people affected by dementia live at home..." why did not writing as "most patients with dementia at home are needed to help...." Un-paid instrumental support is "informal caregiver" or not.

Should be clear what terms are used in qualitative study, do not use quantitative term as "intervention" and "effect" "measure". Please revise all terms are appeared in the manuscript and then used the appropriate term.

Introduction

Line 35-74 be clear what really main problem statement that you try to explore.

Line 76-80 be clear what main objectives are. I found, it is very general terms and not reflect to purpose of study.

Methods

Line 83-87 should be extent what main design did you apply in this study.

Line 100-115 not clear what procedures.

* moved ethical issues into the last page and then should be identified, what ID this work has the approval of ethical consideration.

Finding

Quotations are very difficult to read. Should be revised and make short terms and reflection to participant statement.

Discussion

Line 470-481 not clear. Should be revised and then focusing on main findings and discussed with other scholars.

The discussion appear to be strong. I suggested you add some brief support to clarify the many strengths of this study as follows:

Otobe, Y., Kimura, Y., Suzuki, M. et al. Factors Associated with Increased Caregiver Burden of Informal Caregivers during the COVID-19 Pandemic in Japan. J Nutr Health Aging 26, 157–160 (2022). https://doi.org/10.1007/s12603-022-1730-y

Francisco Javier Rosas-Santiago, Janet Jiménez Genchi, Isaí Sotelo Heredia & Víctor Enrique Ramírez Zamora (2022) Psychoeducation and group acceptance and commitment therapy as psychological support strategies for informal caregivers of patients with a first psychotic episode: an experimental study, Psychosis, DOI: 10.1080/17522439.2022.2061041

Mamom, J.; Daovisan, H. Listening to Caregivers’ Voices: The Informal Family Caregiver Burden of Caring for Chronically Ill Bedridden Elderly Patients. Int. J. Environ. Res. Public Health2022, 19, 567. https://doi.org/10.3390/ijerph19010567

Boumans, J., van Boekel, L. C., Verbiest, M. E., Baan, C. A., & Luijkx, K. G. (2022). Exploring how residential care facilities can enhance the autonomy of people with dementia and improve informal care. Dementia, 21(1), 136–152. https://doi.org/10.1177/14713012211030501

Lindsay Groenvynck, Bram de Boer, Audrey Beaulen, Erica de Vries, Jan P H Hamers, Theo van Achterberg, Erik van Rossum, Chandni Khemai, Judith M M Meijers, Hilde Verbeek, The paradoxes experienced by informal caregivers of people with dementia during the transition from home to a nursing home, Age and Ageing, Volume 51, Issue 2, February 2022, afab241, https://doi.org/10.1093/ageing/afab241

4.1 Barriers and benefits in seeking help. This sub-section is not applicable for separating and discussing.

The conclusion is too short. Should be extended and modified what really main findings are.

Many grammar errors and sentences are invalid meaning. Should be provide native speaker proofread/edit is required.

Author Response

Dear reviewer,

Thank you very much for your precious time and feedback, please see the attachment below. 

Best wishes,

Anna Messina and co-authors

Reviewer 2 Report

Thanks for asking me to review this important topic.

The background provides a good summary of research thus far.

Methods: QA should be referenced. 

3.1 Detail how many attended each of the 5 focus groups. 

Author Response

(The authors gave the same response as above.)

Reviewer 3 Report

I had a couple of main issues with this article, mainly relating to the role of iSupport. If you take out iSupport you haven't really found anything new from your study, which makes me wonder what I get out of it as a reader. If the novelty aspect is iSupport, there's nothing really in what you report that makes me see how your findings relate to it. 

Please see my comments below for more detail.

Comments about iSupport

I’m confused by the mention of iSupport. There is no mention of it in the article title, so when you introduce it in the abstract it’s a bit of a surprise. Then the abstract findings don’t actually appear to explore barriers/facilitators in relation to it, but it’s randomly tagged onto the end like an afterthought.

I’m generally confused about the role/importance of iSupport throughout the study and article. I can’t work out if it’s actually underpinning your study, or if you’re seeing how your findings could relate to it. In some cases it seems like a minor part as there is one very brief line suggesting that you’re doing a study to find out attitudes/barriers/facilitators ‘in view of tailoring iSupport’. However, your topic guide was developed ‘according to the WHO iSupport adaptation guidelines’ which suggests that it’s a more fundamental part of your study, implying that you wanted to tailor iSupport to enable it to be used in Switzerland, and in order to do this you had to find out attitudes/barriers/facilitators. It’s a subtle difference, but it actually makes a big difference in terms of your approach. Tailoring iSupport is not a bi-product of your study (as it initially appears), the purpose of your study is to enable iSupport to be tailored.

The study design says about collecting experiences and attitudes towards iSupport, but apart from being given a presentation about it, there’s nothing to say that any participants would have experience/knowledge of it – unless they happened to be identified through the iSupport community advisory board. If someone’s only knowledge is a ‘detailed presentation of the iSupport project’ could that potentially influence their responses/views?

Also, the study procedure says one area of discussion was ‘attitudes towards support and training programs’ – I assume this covers/includes iSupport – but there’s not enough detail to know what prompts were used as part of this discussion. What questions did you ask about iSupport? I’m also not sure where the results relate to attitudes towards training programs.

The first bit in the results section about the structure doesn’t mention iSupport, so I’m not sure where it fits. More generally, how does iSupport relate to what the caregivers were reporting? If it’s online education, problem solving, psychoeducation and relaxation, how does that relate to peer support, helping people feel free to speak, not judged, giving respite through the person going to a day centre or getting someone to come in and help? What did caregivers say about accessing online support or education? What did they say about iSupport? Were they even asked about it?

It feels like you mention iSupport in section 4.4 as an aside or afterthought, such as making a point along the lines of ‘this is what needs to be done by everything, including iSupport’ or ‘this needs to happen in these cases, including iSupport’. I really don’t know how what you found will help to tailor iSupport. If your findings were ‘highly consistent with previous evidence’ how does that mean that iSupport should be tailored for local needs? Surely if you’re finding the same issues as elsewhere, tailoring isn’t needed?

You also talk about ‘accepting and using iSupport’ but I can’t see anything that actually relates to this. What did people say about it, because there’s no mention of iSupport in your findings.

I suppose my overall point is: you need to be clearer about how and where iSupport relates to your study and your findings. At present, it almost feels like if you took out any mention of iSupport it wouldn’t have any impact on your article. You’ve got your study/findings, then you’ve got iSupport, and they are pretty much separate. If I’m missing something I apologise, but I’m really struggling to see the connection between the two and don’t feel that lines 602-618 are sufficient – especially when they include vague phrases like ‘these considerations seem to apply to how iSupport is locally adapted and implemented…’

Other comments

I think you could be a bit more consistent with your terminology as you seem to alternate freely between family caregivers and informal caregivers, when I would say that they are not necessarily the same thing. Also, you use family caregivers in the article title but have informal caregivers as a keyword, which just seems odd. You also just use caregivers at times, and I’m not clear if you still mean family caregivers or caregivers more generally (e.g. including professional/paid carers).

In the results when you talk about ‘several caregivers’ (e.g. line 167), it would be useful to know how many. When you’re referring to a group of 13, ‘several’ could actually be significant.

Lines 181-182 – it could do with a slight rewording to make it read better, such as ‘less burdened a few months after her father’s diagnosis and therefore being willing to’

I think you need to check the use of he/she and him/her in a few places, as it can be confusing to read. E.g. on line 192 you give an example of a participant ‘considering himself’ but follow it with a quote from a daughter. Are they the same or different participants? If the same, surely it should be himself/son or herself/daughter. If different, this needs to be clearer. (I accept this might be a language/translation issue, but it would be good to pay attention to this to make it more readable and less distracting)

I think that there are limitations that you haven’t covered. You mention gender briefly as a limitation, but don’t actually say why it could be an issue that you only included three men. Could men have different experiences/views, which in turn could have an impact on how to tailor iSupport? If you’re mainly basing your findings on women and use this to tailor iSupport, could you be making it less appropriate for men?

Also, you don’t seem to consider your group size. You only contacted 20 people to be involved (which doesn’t seem many considering how you tried to recruit), and ended up with 13. Although I appreciate this is a ‘convenient sample’, it seems very small when you had five focus groups. This is only 2-3 per group on average (or were some focus groups essentially an interview with one person and others had five people in? there’s no information about this), and I wonder what the impact of this could have been. It could be a positive as it might be less scary for some people, so they might have contributed more. Conversely, it could be a negative as people could feel more on the spot in a small group as there is nowhere to hide, and you don’t get the same level of bouncing ideas off each other or having thoughts triggered by what others say. There should be at least some acknowledgement of the potential issues this could cause.

Author Response

Dear reviewer,

Thank you very much for your precious time and feedback. Please see the attachment below,

Best regards,

Anna Messina and all co-authors

Round 2

Reviewer 1 Report

All comments are revised and well-suited for publication.